# Musculoskeletal pain among desk-based officials of Bangladesh: Association with mental health and individual factors

**Asif Iqbal Ahmed**[1‡], **Shahriar Hasan**[1,2‡], **Md Shahjalal** [1,2]*, **Rony Shaha**[2,3], **Mohammad Delwer Hossain Hawlader**[1], **Mohammad Morshad Alam**[2,4]

**1** Department of Public Health, North South University, Dhaka, Bangladesh, **2** Research Rats, Dhaka, Bangladesh, **3** Department of Microbiology, Noakhali Science and Technology University, Noakhali, Bangladesh, **4** HNP Global Practice, The World Bank, Bangladesh Office, Dhaka, Bangladesh

‡ These authors share first authorship on this work.
* md.shahjalal3@northsouth.edu

**Data Availability Statement:** Data are available at: https://osf.io/skv46/files/osfstorage.

## Abstract

Musculoskeletal (MS) pain is widely prevalent and is an important health issue for desk-based employees which has a negative impact on both personal and work life. This study aimed to determine the MS pain status and its association with mental health and other individual factors among desk-based officials of Dhaka, Bangladesh. This cross-sectional study comprised a sample of 526 desk-based officials from Dhaka, Bangladesh. Data were collected between November 2020 to March 2021. MS pain was determined by the visual analog scale (VAS) and depression and anxiety were screened by Hospital Anxiety and Depression Scale (HADS). Logistic regression analyses were employed to estimate the adjusted effect of independent factors on MS pain. The overall prevalence of MS pain was 64% among desk-based officials. The corresponding prevalence were 19% severe, 21% moderate and 24% mild MS pain. In the adjusted model, gender (AOR: 0.19, 95% CI: 0.07–0.46), BMI (AOR: 0.28, 95% CI: 0.14–0.59), monthly income (AOR: 5.17, 95% CI: 2.18–12.25), organization type (AOR: 4.3, 95% CI:1.8–10.1), floor living (AOR: 4.7, 95% CI:2.1–10.8), physical activity (AOR: 0.16, 95% CI: 0.06–0.45), and lift facility in the house (AOR: 4.11, 95% CI: 2.06–8.23) were associated with MS pain. In addition, the prevalence of anxiety and depression was 17.7% and 16.4%, respectively. Depression was identified as a significant predictor for severe MS pain (AOR: 2.44, 95%CI:1.29–4.63). This study has revealed a relatively high prevalence of MS pain and mental health problems among Bangladeshi desk-based officials. Preventive measures need to be taken from both organizational and personal sides to delimitate MS pain and mental health problems.

## Introduction

Physical and mental health are closely related and influence one another in many ways [1]. There is a psychological dimension to all physical health problems, especially when they involve learning to live with long-term conditions [2]. Musculoskeletal (MS) disorders are

**Funding:** The authors received no specific funding for this work.

**Competing interests:** The authors have declared that no competing interests exist.

injuries or pains in the muscles, bones, nerves, joints, ligaments, and blood vessels as well as one of the major causes of the disease burden [3]. Pain in the neck, shoulders and low back are highly correlated and often occurs together known as MS pain [4] and this pain may result from repetitive motions, forces, rapid work pace, repeated or prolonged activity and vibration [3]. For instance, employees who spend long periods of time at their desks during the course of their work day are more likely to develop MS pain symptoms [3, 5, 6].

Scientific reports indicated that the most common MS pain complaints in the working population are neck pain, shoulder pain, arm pain and back pain etc. [7–9]. For example, 24% reported low back pain, 20% shoulder pain, 21% knee pain, and 9% wrist pain among twelve occupational groups in the United Kingdom [10], 28% low back pain, 21% neck pain, 17% shoulder pain among workers in Japan [11], 40% neck pain, 37% shoulder and 36% low back pain bankers in Pakistan [12], 81.9% back pain, 38.6% neck, 6.8% wrist, and 15.2% shoulder bank employee in Punjab, India [13], and 86.3 neck pain, 75.5% low back pain among computer using office worker in China [14], 17.9% shoulder pain, 15.2% lower back pain, 13.8% neck pain, and 10.8% knee pain among garment workers in Bangladesh [15].

A growing number of the literature suggests that the working environment is not the sole factor affecting the development of MS pain; factors such as gender, age, and body mass index (BMI) play a significant role [5, 16–18].The American Productivity Audit (APA) indicated that MS pain conditions have a profound impact on the ability to work and productivity [18]. In addition, there is strong evidence that MS pain can also be associated with psychological risk factors [5, 19–21]. A systematic review documented that psychological variable such as stress, distress or anxiety as well as mood and emotions were related to the onset of pain, and to acute, subacute, and chronic pain [20]. A prospective study conducted in the UK indicated that increased risks of onset were associated with high levels of psychological distress [2]. Such as, MS pain leads to reduced work ability, functional capacity limitations, frequent sickness absences, chronic disability, early retirement, and impaired quality of life [5, 18–21].

In recent years, a growing number of studies were conducted in Bangladesh on MS disorders/pain with different study participants and locations, where two studies were conducted among garment workers [15, 22–25], and a study was conducted among rural communities [23]. The most recent two studies were conducted among the general Bangladeshi population and Bangladeshi adults; they investigated the association between leisure and MS pain during the pandemic [24] and the prevalence of MS conditions and related disabilities in the adult population [25].

To our knowledge, these studies did not look at desk-based office workers who are more prone to MS pain and psychological consequences. By exploring the prevalence, associated risk factors and psychological impact on MS pain, we can tackle the work absenteeism, lower productivity and mental health problems among office-based workers in Bangladesh. Therefore, this current study seeks to investigate association of physical and mental health and individual factors associated with MS pain in desk-based officials.

## Methods

### Study design and participants

A cross-sectional study was conducted among desk-based officials in Dhaka, Bangladesh. Dhaka is Bangladesh's capital and largest city, with two city corporations, the North and the South [24]. The sample was collected from conveniently selected these two-city corporations from November 2020 to March 2021. We arranged an interview if there was a desk-based officer of the household with at least one-year of experience. However, we excluded subjects under psychiatric treatment and refused to give written consent.

The sample size was calculated based on the assumptions of an alpha of 0.05, a confidence interval of 95% and a prevalence of musculoskeletal condition of 30.4% in Bangladesh [23]. The sample size thus calculated was 345. After adding a 20% non- response rate the final sample size was calculated to be 414. However, we collected in total of 526 data to represent the population better, avoid sampling error and have more accuracy of the result. The process of data selection, identification and inclusion flow diagram is presented in **S1 File**. Data were collected using a validated English version self-administrated questionnaire through face to face interview from study participants (**S2 File**). Before participants filled out the questionnaires, data collectors explained the study's purpose and clarified that participation was voluntary. After the collection of data, an MS excel sheet was created. All data were managed in the MS excel sheet; after that, Stata version 14 were used for analytical exploration. Descriptive statistics of demographic characteristics and frequency of exposure to musculoskeletal pain were calculated. In the analytical portion, multinomial logistic regression analysis was used to determine which socio-demographic characteristics (including age, gender, income, BMI, organization type, floor living, physical activity/day, weekly working time, sedentary activities, lift facility in house, level of anxiety and level of depression) associated with having MS pain. Adjusted regression analysis was performed based on individual explanatory variables only. In the adjusted final model, only potential explanatory variables were considered if their covariate had a label that showed statistical significance with a p-value of less than 5% in the chi square model.

## Measures

### Independent variables

Independent variables included participants' demographic characteristics, order of the floor, physical activity, weekly working time, sedentary activities, lift facility in the house, level of anxiety and level of depression. Age group (<35 years, 35–54 years and >54 years), gender (male vs female), height (inches), weight (kilogram), monthly income in Bangladeshi Taka (BDT) (≤35000 BDT (about 340 US dollar), 35000–60000 BDT and >60000 BDT), organization (government vs private) were among the sociodemographic characteristics. The order of the floor where they lived in were measured by floor of living (≤2nd floor, 3rd-6th floor and ≥ 7th floor). Physical activity per day has 3 categories which were (no, <30 minutes and ≥ 30 minutes). Weekly working time (≤40 hours, 41–47 hours and ≥ 48 per week) and sedentary activities (< 2 hours, 2–4 hours and > 4 hours per day) also have 3 categories subsequently. The lift facility was measured by whether they had a lift in their house or not. The level of anxiety and depression were measured in three categories (normal, border and case).

### Outcome variable

The survey outcome variable was the MS pain of the human body. This study considered only the upper extremities MS pain as the main outcome variable. To measure the MS pain, visual analog scale (VAS) was used [26]. This scale has four categories which are no pain, mild pain, moderate pain, and severe pain. To perform the unadjusted and adjusted logistic regression model and chi square, the category of the outcome variable was divided into three groups: Severe, Mild to Moderate, and No. In this study, we also measured participants anxiety and depression. To screen the anxiety and depression, Hospital Anxiety and Depression Scale (HADS) was used. Using this scale, we categorized anxiety and depression into three categories which are "normal" (score 0–7), "borderline anxiety/depression" (score 8–10) and "clinical anxiety/depression" (score 11–21) [27].

### Ethical considerations

The written informed consent was obtained from each participant. Participants were assured that their personal information would remain confidential and be used only for academic purposes. In addition, participants were informed that they could withdraw at any time without negative consequences. The study protocol was approved by the Institutional Review Board (IRB) of North South University, Dhaka, Bangladesh (2020/OR-NSU/IRB NO.0901).

## Result

The total sample consisted of 526 desk-based workers. The respondents were aged between 24 to 65 years old and the mean age was 39.58±10.5 (SD) and the median age was 36 years. Among the males the mean age was around 41.4±10.9 (SD) years and for females it was 35.2 ±8.1 (SD) years. The mean BMI of the study respondents was 24.8±2.6 (SD) kg/m$^2$ (Male: 24.8 kg/m$^2$; Female: 24.6 kg/m$^2$) with a maximum of 33.6 kg/m$^2$ and a minimum of 18.7 kg/m$^2$. The maximum monthly income was found to be 300,000 BDT with a mean 74,047±57,524 (SD) BDT (male: 74,470 BDT, female: 73,044 BDT) (**S3 File**).

In this study we found that about 19% of the participants had severe MS pain, 21% had moderate MS pain and 24% had mild MS pain as well (**Fig 1**).

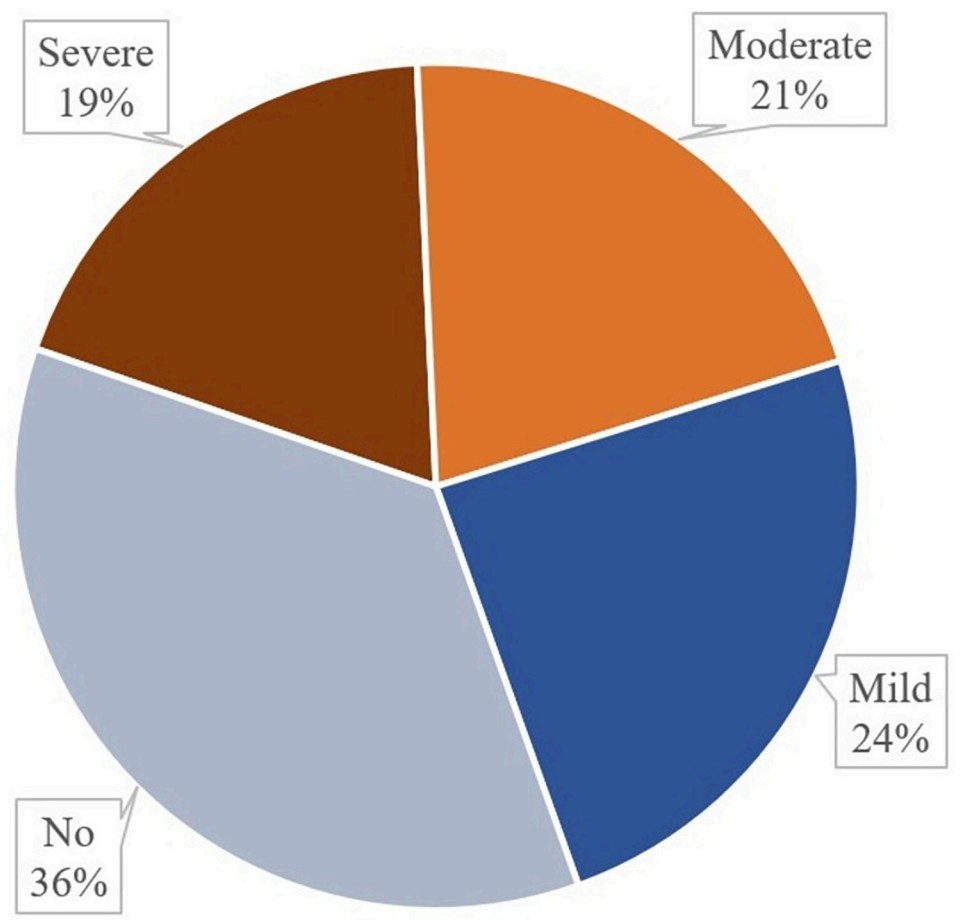

**Fig 1. Prevalence of musculoskeletal pain among the participants.**

**Table 1. Association of socio-demographic characteristics and activity related characteristics with MS pain.**

| Variables | Category | Musculoskeletal Pain | | | p-value |
|---|---|---|---|---|---|
| | | Severe (%) | Mild to Moderate (%) | No (%) | |
| Age | < 35 years | 25 (12.32) | 96 (47.29) | 82 (40.39) | **0.007** |
| | 35–54 years | 49 (20.68) | 106 (44.73) | 82 (34.60) | |
| | > 54 years | 26 (30.23) | 36 (41.86) | 24 (27.91) | |
| Gender | Male | 62 (16.76) | 156 (42.16) | 152 (41.08) | **<0.01** |
| | Female | 38 (24.36) | 82 (52.56) | 36 (23.08) | |
| Income | ≤35000 BDT | 29 (18.59) | 60 (38.46) | 67 (42.95) | **<0.01** |
| | 35000–60000 BDT | 15 (11.72) | 87 (67.97) | 26 (20.31) | |
| | > 60000 BDT | 56 (23.14) | 91 (37.60) | 95 (39.26) | |
| BMI | Obese | 62 (23.85) | 94 (36.15) | 104 (40.0) | **<0.01** |
| | Overweight | 17 (12.69) | 80 (59.70) | 37 (27.61) | |
| | Normal | 21 (15.91) | 64 (48.48) | 47 (35.61) | |
| Type of organization | Government | 37 (12.94) | 155 (54.20) | 94 (32.87) | **<0.01** |
| | Private | 63 (26.25) | 83 (34.58) | 94 (39.17) | |
| Floor living | ≤ 2nd floor | 27 (13.2) | 85 (41.7) | 92 45.1%) | **<0.01** |
| | 3rd to 6th floor | 68 (28.2%) | 119 (49.4) | 54 (22.4) | |
| | ≥ 7th floor | 5 (6.2) | 34 (42.0) | 42 (51.9) | |
| Physical activity/day | No | 46 (25.27) | 96 (52.75) | 40 (21.98) | **<0.01** |
| | < 30 minutes | 49 (20.33) | 74 (30.71) | 118 (48.96) | |
| | ≥ 30 minutes | 5 (4.85) | 68 (66.02) | 30 (29.13) | |
| Weekly working time | ≤ 40 hours | 17 (10.69) | 86 (54.09) | 56 (35.22) | 0.002 |
| | 41–47 hours | 55 (20.83) | 105 (39.77) | 104 (39.39) | |
| | ≥ 48 hours | 28 (27.18) | 47 (45.63) | 28 (27.18) | |
| Sedentary activities | < 2 hours | 23 (13.8) | 74 (44.3) | 70 (41.9) | **<0.01** |
| | 2–4 hours | 16 (12.5) | 48 (37.5) | 64 (50.0) | |
| | > 4 hours | 61 (26.4) | 116 (50.2) | 54 (23.4) | |
| Lift facility in house | Yes | 51 (19.4) | 132 (50.2) | 80 (30.4) | 0.029 |
| | No | 49 (18.6) | 106 (40.3) | 108 (41.1) | |

We also found that MS pain was significantly associated with age (p = 0.007), gender (p<0.01), monthly income (p<0.01), BMI (p<0.01), type of the organization (p<0.01), the floor where live in (p<0.01), physical activity per day (p<0.01), weekly working time (p = 0.002), sedentary activities (p<0.01), lift facility in the house (p = 0.029) (**Table 1**).

**Table 2** shows the association of sociodemographic characteristics and MS pain. We found that respondents who were more than 54 years of old faced 2.21 times severe (AOR = 2.21; 95% CI: 0.66–7.3, p = 0.19) MS pain than the respondents who were less than 35 years. At the same time, the respondents from the 35 to 54 years age group also felt severe pain (AOR = 2.09; 95% CI: 0.92–4.8, p = 0.08) than the younger aged (<35 years old) respondents. On the other hand, the male respondents felt less severe pain (AOR = 0.19; 95% CI: 0.07–0.46, p = <0.01) rather than the females. The respondents who earned >60,000 BDT/month had less likely to chances (AOR = 0.51; 95% CI:0.15–1.76, p = 0.29) to have MS pain than who earned ≤35000 BDT/month. The overweight respondents felt 1.23 times severe pain (AOR = 1.23; 95% CI: 0.42–3.59, p = 0.69) compared to the respondents who had normal weight. Besides, the private employees felt 4.3 times severe pain (AOR = 4.3; 95% CI: 1.8–10.1, p = <0.01) than the government employees. This study also found that, the respondents who lived between 3rd and 6th floor, felt 4.7 times severe pain

**Table 2. Logistic regression analysis of sociodemographic characteristics and musculoskeletal pain (multinomial logistic regression).**

| Variables | Category | | | | Musculoskeletal Pain | |
|---|---|---|---|---|---|---|
| | | | | | Severe vs No | Mild to Moderate vs No |
| | | | | | Adjusted model | Adjusted model |
| | Label | Severe | Mild to Moderate | No | OR (95% CI) | OR (95% CI) |
| Age | > 54 years | 26 (30.23) | 36 (41.86) | 24 (27.91) | 2.21 (0.66–7.3) | 0.97 (0.3–2.6) |
| | 35–54 years | 49 (20.68) | 106 (44.73) | 82 (34.60) | 2.09 (0.92–4.8) | 0.97 (0.51–1.83) |
| | < 35 years | 25 (12.32) | 96 (47.29) | 82 (40.39) | Reference | |
| Gender | Male | 62 (16.76) | 156 (42.16) | 152 (41.08) | 0.19 (0.07–0.46) | 0.29 (0.14–0.60) |
| | Female | 38 (24.36) | 82 (52.56) | 36 (23.08) | Reference | |
| Income | > 60000 BDT | 56 (23.14) | 91 (37.60) | 95 (39.26) | 0.51 (0.15–1.76) | 0.56 (0.23–1.36) |
| | 35000–60000 BDT | 15 (11.72) | 87 (67.97) | 26 (20.31) | 0.84 (0.26–2.69) | 5.17 (2.18–12.25) |
| | ≤35000 BDT | 29 (18.59) | 60 (38.46) | 67 (42.95) | Reference | |
| BMI | Obese | 62 (23.85) | 94 (36.15) | 104 (40.0) | 0.64 (0.27–1.55) | 0.28 (0.14–0.59) |
| | Overweight | 17 (12.69) | 80 (59.70) | 37 (27.61) | 1.23 (0.42–3.59) | 2.7 (1.19–6.3) |
| | Normal | 21 (15.91) | 64 (48.48) | 47 (35.61) | Reference | |
| Organization type | Private | 63 (26.25) | 83 (34.58) | 94 (39.17) | 4.3 (1.8–10.1) | 1.5 (0.76–2.9) |
| | Government | 37 (12.94) | 155 (54.20) | 94 (32.87) | Reference | |
| Floor living | ≥ 7th floor | 5 (6.2) | 34 (42.0) | 42 (51.9) | 0.25 (0.06–1.0) | 0.21 (0.08–0.54) |
| | 3rd to 6th floor | 68 (28.2%) | 119 (49.4) | 54 (22.4) | 4.7 (2.1–10.8) | 1.4 (0.72–3.04) |
| | ≤ 2nd floor | 27 (13.2) | 85 (41.7) | 92 (45.1%) | Reference | |
| Physical activity/day | ≥ 30 minutes | 5 (4.85) | 68 (66.02) | 30 (29.13) | 0.20 (0.04–0.84) | 0.62 (0.24–1.5) |
| | < 30 minutes | 49 (20.33) | 74 (30.71) | 118 (48.96) | 0.16 (0.06–0.45) | 0.11 (0.04–0.27) |
| | No | 46 (25.27) | 96 (52.75) | 40 (21.98) | Reference | |
| Weekly working time | ≥ 48 hours | 28 (27.18) | 47 (45.63) | 28 (27.18) | 1.85 (0.68–5.0) | 0.80 (0.35–1.8) |
| | 41–47 hours | 55 (20.83) | 105 (39.77) | 104 (39.39) | 2.82 (1.21–6.5) | 0.81 (0.44–1.5) |
| | ≤ 40 hours | 17 (10.69) | 86 (54.09) | 56 (35.22) | Reference | |
| Sedentary activities | > 4 hours | 61 (26.4) | 116 (50.2) | 54 (23.4) | 1.2 (0.54–2.8) | 1.22 (0.60–2.45) |
| | 2–4 hours | 16 (12.5) | 48 (37.5) | 64 (50.0) | 0.50 (0.18–1.36) | 0.65 (0.31–1.36) |
| | < 2 hours | 23 (13.8) | 74 (44.3) | 70 (41.9) | Reference | |
| Lift facility in house | Yes | 51 (19.4) | 132 (50.2) | 80 (30.4) | 2.3 (1.0–5.5) | 4.11 (2.06–8.23) |
| | No | 49 (18.6) | 106 (40.3) | 108 (41.1) | Reference | |

(AOR = 4.7; 95% CI: 2.1–10.8, p = <0.01) than the people who lived in less than or equal to 2nd floor.

The respondents who did physical activity ≥30 minutes per day had less likely chances to have severe pain (AOR = 0.20; 95% CI: 0.04–0.84; p = 0.02) than who didn't do any physical activity. The people who worked more ≥48 hours per week, had 1.85 times higher chances (AOR = 1.85; 95% CI: 0.68–5.0, p = 0.22) of having severe MS pain than who worked ≤40 hours per week. The respondents who spent >4 hours on the sedentary activities, had 1.2 times higher chances (AOR = 1.2, 95% CI: 0.54–2.8, p = 0.60) of having severe pain than who spent <2 hours daily on sedentary activities. The respondents who had lift facilities in their house, had 2.3 times higher chances (AOR = 2.3; 95% CI: 1.0–5.5, p = 0.049) of having MS pain than the respondents who had not any lift facilities in their house.

In this study it was found that 17.68% participants had anxiety (case) and 13.88% participants had borderline anxiety. On the other hand, 16.35% participants had depression (case) and 13.12% had borderline depression (Fig 2). This study also found that 4.9% of participants had both anxiety and depression.

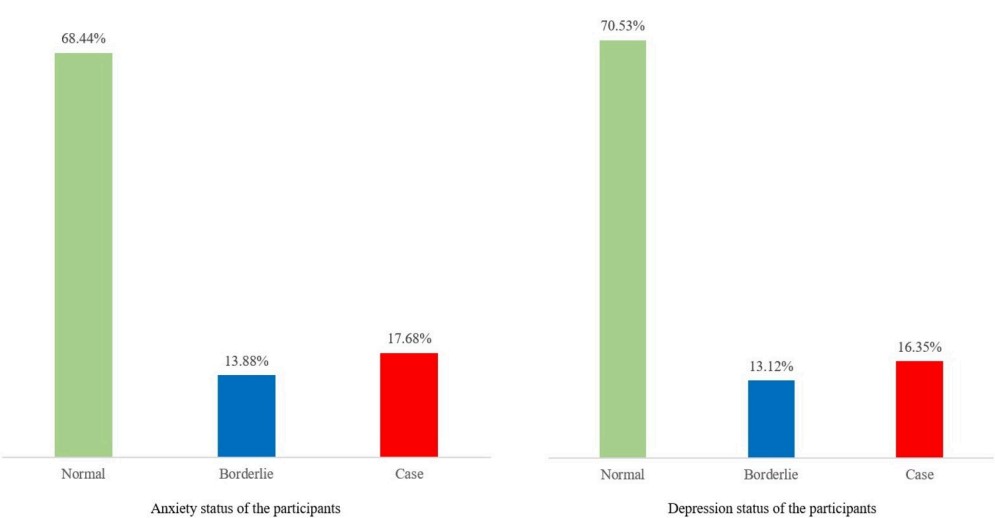

**Fig 2. Anxiety and depression status of the participants.**

## Association of depression and anxiety and MS pain

Table 3 shows the level of depression was found significantly associated with MS pain. The respondents who had borderline depression, had 7.81 times higher chances (AOR = 7.81; 95% CI:3.71–16.46, p = <0.01) of having severe MS pain than the respondents who didn't have any depression. Besides the respondents who had borderline anxiety, they also had 1.15 times higher chances (AOR = 1.15; 95% CI: 0.56–2.37, p = 0.695) of having severe MS pain than the respondents who had no anxiety at all.

## Discussion

This study demonstrated the association of physical and mental health and individual factors associated with MS pain in desk-based officials in Dhaka, Bangladesh. We found that MS pain is significantly associated with selected socio demographic variables including age, gender, monthly income, BMI, type of employment, the floor where live in, physical activity, weekly working time, sedentary activities and lift facility in the house. Similar with this, Kalinieni et al. (2016) found out that age, gender, computer work experience and BMI were significantly associated with MS pain [5].

According to our data, we found that respondents who were aged more than 54 years, were 2.21 times very irritated about their MS pain than the respondents who were aged less than 35

**Table 3. Logistic regression analysis with anxiety and depression with musculoskeletal pain (multinomial logistic regression).**

| Variables | Category | | | | Musculoskeletal Pain | |
|---|---|---|---|---|---|---|
| | | | | | Severe vs No | Mild to Moderate vs No |
| | | | | | Adjusted model | Adjusted model |
| | | Severe | Mild to Moderate | No | OR (95% CI) | OR (95% CI) |
| Level of Anxiety | Case | 20 (21.51) | 44 (47.31) | 29 (31.18) | 0.90 (0.45–1.79) | 1.23 (0.71–2.11) |
| | Border | 15 (20.55) | 30 (41.10) | 28 (38.36) | 1.15 (0.56–2.37) | 0.85 (0.48–1.50) |
| | Normal | 65 (18.06) | 164 (45.56) | 131 (36.39) | Reference | |
| Level of Depression | Case | 24 (27.91) | 30 (34.88) | 32 (37.21) | 2.44 (1.29–4.63) | 0.69 (0.40–1.21) |
| | Border | 31 (44.93) | 25 (36.23) | 13 (18.84) | 7.81 (3.71–16.45) | 1.41 (0.69–2.90) |
| | Normal | 45 (12.13) | 183 (49.33) | 143 (38.54) | Reference | |

years. Similar with this finding, a recent systematic review has found that among the working population shoulder pain continued to increase after the age of 50 years [28]. It might be due to normal physical condition, for example, older aged people have a lower ability to tolerate pain. Another possible reason could be aging affects brain functioning [29]. Future research needs to explore how pain perception changes in older people.

Cavallari et al. (2016) showed that working female respondents were facing higher MS pain than their male counterpart regardless of the job responsibilities they have to be performed [30]. This finding in line with our result where we found that male respondents were less likely irritated about the MS pain rather than the females. The possible reason could be the risk factors at work such as repetitive work, poor ergonomic equipment as well as the risk factors at home as women have less opportunity to relax at home [30].

Our result indicated that respondents who earned more than 60000 BDT per month had less likely chances to have MS pain than the respondents who earned less than or equal to 35000 BDT per month. Similar with this findings Rutz and Doner (2017) found that individual with lower socio-economic status was more prone to poor psychosocial and poor working conditions which lead to higher prevalence of severe MS pain and sickness among the lower income group in Australia [31]. It could be due to economic insecurity, or when people are in a mental state of financial insecurity, regardless of how it was induced, they experience greater physical pain [32]. A previous study conducted in a Portuguese specialized obesity clinic in 2021 found that all obesity indicators were associated with MS pain [33]. Similarly, this present study found that the overweight respondents were significantly more bothered about the MS pain than the respondents who had normal BMI. It could be due to the obesity increase mechanical stress in the body which in turn increase the muscles and joints pain [34].

The findings also showed that the respondents, who had borderline depression, had more than seven times higher chances of having MS pain than the respondents who didn't have any depression. Besides the respondents who had borderline anxiety, they also had higher chances of having MS pain than the respondents who didn't have any anxiety at all. In line with this, Ebrahimzadeh et al. (2019) highlighted that prevalence of depression and anxiety increased pain and limb disability [35]. The reason could be that anxiety and depression negatively affect the thought which hinders rehabilitation from any disease and also increase pain [36].

The present study reflects that physical and mental health are closely related with each other and also existence of association of this health with MS pain specially among the desk-based officials. However, these MS pain or musculoskeletal problems have long time consequences in human life including prolonged morbidity, disability and work loss etc. [37]. These problems clearly suggest that there is a need of taking proper initiatives to solve them immediately. By finding out the association between MS pain and work ergonomics, Kaliniene et al (2016) recommended that there should take some preventative measures in workplace including improvement in work environment, education and workload optimization is needed [5]. In line with these suggestions, responsible authorities need to pay attention to alleviate these burdens by ensuring workload optimization and a sound and safe workplace for desk-based officials and/or human resources.

Even though a growing number of literatures worked with the risk factors, patterns and impact of MS pain, a little research has done considered the desk-based office workers who are more prone to MS pain and psychological consequences in Bangladesh. Therefore, more emphasis should be placed on interventions targeting desk-based office workers. We are not first to highlighting that physical and mental health is associated with MS pain especially among the desk-based officials. Therefore, it is pertinent to mention that our findings of this study will be helpful to take proper initiative to palliate MS pain among Bangladeshi office-based officials and their mental well-being.

This study has some limitations. Our study has considered selected socio demographic variables and activity related characteristics which are limited in number. Considering more variables would be helpful to predict the risk factors more precisely. This study also has geographical limitation as we collected data from one city. Sample from different cities could be change the findings. Based on these limitations, this study recommend that researchers of occupational health sciences should perform more research on this issue by considering an immense number of risk factors and geographical variations.

## Conclusion

The study found one in five study participants had MS pain. MS pain was significantly associated with sociodemographic and lifestyle factors as well as mental health among desk-based officials in Dhaka, Bangladesh. It is important recognize that a considerable number of desk-based officials are suffered from physical and mental health problems in Bangladesh. The suitable preventive measures need to be implemented in offices as well as ensure mental health counselling. Future interventions may include large sample size and different geographical area.

## Supporting information

**S1 File. Flow chart of participants inclusion for analysis.**
(DOCX)

**S2 File. Questionnaire on MS pain among desk-based officials of Bangladesh: Association with mental health and individual factors.**
(DOCX)

**S3 File. Descriptive statistics of sociodemographic characteristics of the desk-based officials of Dhaka city.**
(DOCX)

## Acknowledgments

The manuscript arose from the first author (AIH) thesis of Master of Public Health (MPH) of North South University, Dhaka, Bangladesh. The authors are thankful to all participants for their support.

## Author Contributions

**Conceptualization:** Asif Iqbal Ahmed, Shahriar Hasan, Md Shahjalal, Mohammad Morshad Alam.

**Data curation:** Asif Iqbal Ahmed, Shahriar Hasan, Rony Shaha, Mohammad Morshad Alam.

**Formal analysis:** Asif Iqbal Ahmed, Shahriar Hasan, Mohammad Morshad Alam.

**Investigation:** Md Shahjalal, Mohammad Delwer Hossain Hawlader, Mohammad Morshad Alam.

**Methodology:** Asif Iqbal Ahmed, Shahriar Hasan, Md Shahjalal, Mohammad Morshad Alam.

**Project administration:** Md Shahjalal, Mohammad Delwer Hossain Hawlader.

**Resources:** Asif Iqbal Ahmed, Rony Shaha, Mohammad Delwer Hossain Hawlader, Mohammad Morshad Alam.

**Software:** Shahriar Hasan, Rony Shaha.

**Supervision:** Mohammad Delwer Hossain Hawlader, Mohammad Morshad Alam.

**Validation:** Shahriar Hasan, Md Shahjalal, Rony Shaha, Mohammad Morshad Alam.

**Visualization:** Shahriar Hasan, Md Shahjalal, Rony Shaha, Mohammad Delwer Hossain Hawlader.

**Writing – original draft:** Asif Iqbal Ahmed, Shahriar Hasan, Md Shahjalal, Mohammad Morshad Alam.

**Writing – review & editing:** Asif Iqbal Ahmed, Shahriar Hasan, Md Shahjalal, Rony Shaha, Mohammad Delwer Hossain Hawlader, Mohammad Morshad Alam.

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
